# Formamidinium Perovskite Deposition in Ambient Air Environment for Inverted p-i-n Solar Cells

**DOI:** 10.3390/nano14010107

**Published:** 2024-01-02

**Authors:** Nadir Vanni, Riccardo Pò, Paolo Biagini, Gianluca Bravetti, Sonia Carallo, Antonella Giuri, Aurora Rizzo

**Affiliations:** 1Dipartimento di Matematica e Fisica “E. De Giorgi”, Università del Salento, Campus Ecotekne, via Arnesano, 73100 Lecce, Italy; nadir.vanni@nanotec.cnr.it (N.V.);; 2Istituto di Nanotecnologia CNR-NANOTEC c/o Campus Ecotekne, via Monteroni, 73100 Lecce, Italy; sonia.carallo@nanotec.cnr.it (S.C.); antonella.giuri@nanotec.cnr.it (A.G.); 3Renewable Energy, Magnetic Fusion and Material Science Research Center, Istituto Guido Donegani, Eni S.p.A., via Fauser 4, 28100 Novara, Italypaolo.biagini@eni.com (P.B.)

**Keywords:** perovskite, FAPI, inverted perovskite solar cells, ambient air deposition

## Abstract

In order to move towards large-scale fabrication, perovskite solar cells need to detach themselves from strictly controlled environmental conditions and, to this end, fabrication in ambient air is highly desirable. Formamidinium iodide perovskite (FAPI) is one of the most promising perovskites but is also unstable at room temperature, which may make the ambient air deposition more difficult. Herein, we investigated different formulations of pure FAPI for the fabrication of perovskite solar cells (PSCs) in air. We found that formulations using a mixture of N,N-Dimethylformamide (DMF): N-methyl-2-pyrrolidone (NMP) and only dimethyl sulfoxide (DMSO) are suitable for the deposition in air. To fabricate inverted p-i-n solar cells, we tested different hole transporting layers (HTLs) and observed the effects on the wettability of the perovskite solution and on the performance. A self-assembly monolayer of 2PACz (2-(9H-Carbazol-9-yl)ethyl]phosphonic acid) was found to be the best option as a HTL, allowing us to achieve efficiencies >15% on both FTO and ITO.

## 1. Introduction

Since the first appearance of the perovskite material in the photovoltaic (PV) field in 2009 [1] as a dye in nanoporous TiO_2_ for dye-sensitized solar cells, perovskite solar cells (PSCs) have attracted a considerable interest and have grown in power conversion energy (PCE) so much to be compared to the already established PV technologies in just ten years [2]. Starting from the first reported PSC with an efficiency of 10.9% in 2012 [3], the PCEs have been increased to more than 25% [4] in recent years, thanks to the numerous methods to improve the efficiency and stability of the devices, many of which target the passivation of defects that causes ion migration and non-radiative recombination, which ultimately affect the performance and stability of the device [5]. Other strategies include interfacial modifications, acting on the optimization of the charge transport materials [6], and composition engineering [7]. Much of the success of PSCs is due to the excellent optical and electrical properties of the material, such as tunable band gap [8], large optical absorption coefficient [9], and high carrier mobility [10]. Despite the steady progress in both the efficiency and stability of the devices, there are still a number of issues that need to be addressed and resolved in order to successfully upscale manufacturing to a large-scale [11]. One of the first issues on the list is to avoid the strict control of the environmental conditions, usually achieved by an N_2_ filled glovebox, in which the perovskite layer is deposited. Exposure to atmospheric moisture is known to be detrimental for metal halide perovskite materials [12]. This limitation becomes more relevant when moving from lab-scale spin-coater deposition to larger-scale deposition methods, for which having a controlled environment can become quite expensive. For this very reason, it is essential to be able to fabricate PSCs in ambient air conditions, which will also pave the way for the fabrication of other devices such as perovskite and hybrid metasurfaces [13,14], in larger scale. While there are already a number of reports of methylammonium lead iodide (MAPI) PSCs being processed in ambient air, both on a small [15] and large scale, this is not the case for formamidinium lead iodide (FAPI) PSCs. FAPI perovskite is an excellent candidate to replace MAPI in single junction solar cells due to its ideal bandgap of 1.48 eV [16], which is close to the optimal bandgap defined by the Shockley–Queisser limit [17], and also exhibits a longer carrier lifetime [18] and higher thermal stability [19]. However, FAPI perovskite suffers from intrinsic phase instability and is often replaced by mixed cation perovskites, even though the ideal bandgap value is compromised [20]. Due to the larger ionic radius of the FA cation, different polymorphs exist, and they are typically divided into α and δ phases [21]. Of these two, the α-black phase of FAPI is photoactive, however, it is metastable at room temperature and rapidly transitions to the non-perovskite δ-yellow phase [22]. In addition, the phase transition is boosted by exposure to oxygen and moisture, making the deposition and the stabilization in air more complex. Several reports can be found of FAPI PSCs fabricated in a N_2_ controlled atmosphere in an inverted p-i-n configuration, which is the preferred structure in terms of scalability due to the easy fabrication and cost-effectiveness [23], achieving a record PCE of 24.08% [24]. On the contrary, limited cases of FAPI-based p-i-n PSCs fabricated in air have been reported and the increase in efficiency relies on the use of additives in the solution, complicating the overall system [25]. In this context, the behavior of the solution in ambient air and the choice of solvent must be carefully considered, as they are actively involved in the perovskite crystallization process and have a major influence on the final quality of the perovskite sample as well as the stability of the α-phase. Herein, three formulations of FAPI with different solvent systems were investigated for ambient air deposition. Among them, we found that the formulations based on N,N-dimethylformamide (DMF):N-methyl-2-pyrrolidone (NMP) and only dimethyl sulfoxide (DMSO) as the solvent were suitable for ambient air deposition, producing stable FAPI samples with good crystallinity and few defects, as confirmed by the X-ray diffraction (XRD) patterns and steady-state photoluminescence (PL) spectra, and a morphology with compact and large grains. Among these, the DMSO formulation is of particular interest, as it is less toxic when compared to DMF in light of the possible upscaling of perovskite deposition [26]. Furthermore, devices were fabricated with both of the formulations and different hole transporting layers (HTLs) were compared to address the wettability issue that arises in humid ambient air and represents a major challenge, especially for larger-scale depositions [27]. Among the tested HTLs, a monolayer of 2PACz (2-(9H-Carbazol-9-yl)ethyl]phosphonic acid) allowed us to obtain with the DMF:NMP formulation the best PCE of 15.93% on indium tin oxide (ITO) and 15.18% on fluorine-doped tin oxide (FTO).

## 2. Materials and Methods

### 2.1. Chemicals

Lead (II) iodide (PbI_2_, ultradry 99.999% metals basis) was purchased from Alfa Aesar (Kandel, Germany); formamidinium iodide (FAI, >99.99%); 4-fluoro-phenethylammonium iodide (FPEAI) were purchased from GreatCell Solar (Queanbeyan, Australia); methylammonium chloride (MACl); N,N-dimethylformamide (DMF), anhydrous, 99.8%; N-methyl-2-pyrrolidone (NMP), 99.9%; dimethyl sulfoxide (DMSO), anhydrous, ≥99.9%; ethyl acetate (EA), anhydrous, 99.8%; chlorobenzene anhydrous, 99.8% (CB); 2-propanol (IPA); toluene (Tol), anhydrous, 99.8%; PTAA [poly(bis{4-phenyl}{2,4,6-trimethylphenyl}amine)]; bathocuproine, 96% (BCP) were purchased from Sigma Aldrich (St. Louis, MO, USA); 2PACz 2-(9H-Carbazol-9-yl)ethyl]phosphonic acid, >98.0% was purchased from TCI (Chuo-ku, Tokyo, Japan); [6,6]-phenyl C61 butyric acid methyl ester (PCBM) was purchased from Nano-c (Westwood, MA, USA); Poly(3,4-ethylenedioxythiophene) polystyrene sulfonate (PEDOT:PSS) solution (1.3 %wt in water) was purchased form Heraeus (Leverkusen, Germany).

### 2.2. Device Fabrication

The glass ITO patterned substrates (15 × 15 mm^2^) and the glass FTO etched substrates (20 × 20 mm^2^) were sequentially cleaned by ultrasonication in deionized water, acetone, and isopropanol for 10 min each. The ITO and FTO substrates were dried with nitrogen and UV-Ozone treatment was performed before the hole transport layer (HTL) deposition. The PTAA (1.5 mg/mL in toluene) solution was spin coated in ambient air on ITO substrates at 6000 rpm for 30 s and annealed at 100 °C for 10 min. The FPEAI (20 mM in DMF) solution was spin coated in a glove box on the PTAA layer at 4000 rpm for 25 s and annealed at 100 °C for 1 min. The 2PACz (0.5 mg/mL in ethanol) solution was spin coated in a glove box on ITO and FTO substrates at 3000 rpm for 30 s and annealed at 100 °C for 10 min. After cooling them down, the substrates were washed with ethanol to remove the unbound molecules and then annealed again at 100 °C for 5 min [28]. The PEDOT:PSS solution was filtered through PVDF and spin coated in ambient air on FTO substrates at 4000 rpm for 60 s and annealed at 140 °C for 20 min. The FAPbI_3_ solutions were prepared following three formulations. (1) The DMF:NMP solution was prepared by dissolving 461 mg of PbI_2_ (1 mmol), 172 mg of FAI (1 mmol) and 13.5 mg of MACl (0.2 mmol), 96.30 µL of N-Methyl-2-pyrrolidone (NMP, 1 mmol) in N,N-dimethylformamide (DMF, 590.71 µL) and the solution was heated at 60 °C for 1 h. The perovskite solution was spin coated in ambient air at 5000 rpm for 17 s and 400 µL of ethyl acetate was dropped on the spinning substrate at the ninth second. The substrate was annealed for 1 min at 100 °C and 12 min at 165 °C. (2) The DMF:DMSO solution was prepared by dissolving 461 mg of PbI_2_ (1 mmol), 172 mg of FAI (1 mmol), and 13.5 mg of MACl (0.2 mmol) in 1 mL of the mixture of DMF:DMSO (4:1). The perovskite solution was spin coated in ambient air at 4000 rpm for 30 s and 300 µL of ethyl acetate was dropped on the spinning substrate at the tenth second. The substrate was annealed at 150 °C for 10 min. (3) The DMSO solution was prepared by dissolving 461 mg of PbI_2_ (1 mmol), 172 mg of FAI (1 mmol), and 13.5 mg of MACl (0.2 mmol) in dimethyl sulfoxide (DMSO, 1 mL). The perovskite solution was spin coated in ambient air at 4000 rpm for 20 s and 300 µL of ethyl acetate was dropped on the spinning substrate at the eleventh second. The substrate was annealed at 150 °C for 10 min. We deposited about 50 nm of the PCBM layer in a glove box by spin coating 50 μL of filtered solution (25 mg/mL in chlorobenzene) on the active layer at 1000 rpm for 60 s, followed by the deposition of a few nanometers thin layer of BCP by spin coating the solution (0.5 mg/mL in isopropanol) at 6000 rpm for 20 s. In the final step, 100 nm of Ag electrodes was evaporated through a shadow mask in high vacuum. The active area was 0.04 cm^2^ for the ITO substrates and 0.16 cm^2^ for the FTO substrates. The device characterization was performed using a Keithley 2400 Source Measure Unit and Air Mass 1.5 Global (AM1.5G) solar simulator (Newport Corporation, Milano, Italy) exposed to an irradiation intensity of 100 (mW/cm^2^). Current–voltage characteristics of the devices were obtained in the range from 1.2 to 0.2 V.

### 2.3. Characterization

The scanning electron microscopy (SEM) analysis was conducted by a Carl Zeiss Auriga40 Crossbeam instrument (Zeiss, Oberkochen, Germany) in high vacuum and high-resolution mode, equipped with a Gemini column (Zeiss, Oberkochen, Germany) and an integrated high efficiency in-lens detector. A 5 kV voltage acceleration was applied. The film thickness was analyzed using a Dektak 150 optical profilometer (Bruker, Billerica, MA, USA). The XRD patterns of the perovskite films were measured with a PAN analytical X′ Pert-PRO Materials Research Diffractometer (Malvern Panalytical, Worcestershire, UK) using graphite monochromated CuK α radiation (λ = 1.5405 Å). XRD measurements were conducted in ambient condition at room temperature. The photoluminescence (PL) measurements were recorded by means of a Fluorolog^®^—3 spectrofluorometer (HORIBA Jobin-Yvon, Minami-ku, Kyoto, Japan), equipped with a 450 W xenon lamp as the exciting source and double grating excitation and emission monochromators. All of the optical measurements were performed at room temperature on thin film deposited on glass. The PL emission spectra were recorded by using an excitation wavelength of 375 nm. The spectrofluorometer was equipped with a FluoroHub (HORIBA Jobin-Yvon, Minami-ku, Kyoto, Japan) module to perform lifetime measurements by the time-correlated single photon counting (TCSPC) technique. The pulsed excitation source was a laser diode emitting at 485 nm (NanoLED N485L, pulse width <200 ps average power of 14 pJ/pulse) with a repetition rate of 1 MHz. The PL emission was detected by a picosecond photon counter (TBX ps Photon Detection Module, HORIBA Jobin-Yvon, Minami-ku, Kyoto, Japan). The absorption spectrum with and without the top electrode was analyzed by a PerkinElmer Spectrophotometer Lambda 1050 (PerkinElmer, Shelton, CT, USA).

## 3. Results and Discussion

From all the FAPI formulations studied, we selected the following solvent systems: DMF:NMP, DMF:DMSO, and DMSO. The role of the solvent goes far beyond solubilizing the precursors as it participates in the perovskite formation, in particular by controlling the growth of the nuclei and slowing down the reaction rate. Choosing the right solvent therefore becomes essential to achieving a high performance PSC. We chose to test the DMF:DMSO and DMF:NMP formulations as they represent the ideal model of a non-ligand solvent (DMF) and a ligand solvent (DMSO or NMP) with a strong coordination ability to form a stable intermediate adduct, thus retarding the crystallization and allowing for better control and the production of highly uniform FAPI films [29]. While DMSO is the most commonly used additive as a Lewis base [30], it has been shown that NMP is a better option in the case of FA-based perovskites due to the stronger interaction between NMP and the FA cation [31]. Moreover, the toxicity of the solvents also needs be considered, especially looking toward the possible upscaling of the PSC technology. Therefore, we also chose to test a formulation using only DMSO as the solvent to eliminate highly toxic solvents. To test the formulations, all of the perovskite films were deposited in air with uncontrolled relative humidity (RH = 30–70%) and the deposition conditions were adjusted to suit the different formulations. The formulations had different molarities due to the different solubility of the precursors in the solvents, thus the spin-coating parameters were optimized to obtain good quality films with a similar thickness (see the experimental part for details). The optical and morphological characterization of the samples was carried out. It is very clear from all of the evidence in Figure 1 that the DMF:DMSO formulation did not produce FAPI samples of good quality. The absorption spectra are shown in Figure 1a, and while the DMF:NMP and DMSO formulations showed similar absorbance intensity in the range between 850 nm and 600 nm, the DMF:DMSO formulation presented a decrease in absorbance intensity and shoulders at 570 nm and 620 nm, which were caused by the presence of another hexagonal polytype (6H) in the film [32]. In fact, several hexagonal polytypes of lead halide perovskite have been reported, in addition to the pure delta phase, identified as 2H. However, they are all detrimental to the performance and are indicators of the degradation of the film. The XRD patterns of the samples are shown in Figure 1b and the diffraction peaks found at 13.95°, 28.10°, 31.50°, and 42.75° can be assigned to the (1 0 0), (2 0 0), (2 1 0), and (2 2 1) reflections of the cubic α-black phase of FAPI [21,33]. The DMF:DMSO sample showed the lowest crystallinity of the three with the diffraction peak at 12.6°, which can be assigned to PbI_2_ [32] and the α-phase FAPI peak at 13.95° having comparable heights. A small amount of PbI_2_ was also detected in the DMF:NMP and DMSO samples, but in negligible amounts. It is worth noting that the presence of the pure δ-phase FAPI, identified by a peak at 11.80° [34,35], was not detectable in any of the XRD patterns. In comparison, the DMF:NMP sample had a higher crystallinity as the (1 0 0) reflection was higher and sharper when compared to the (1 0 0) reflection in the DMSO sample, which showed a broadening, suggesting a lower crystallinity and smaller grain size [36]. Finally, the low quality of the DMF:DMSO sample was confirmed by the PL spectra in Figure 1c, where the PL intensity of the sample was significantly lower compared to the PL intensity of the DMF:NMP and DMSO samples, whose intensity was comparable, suggesting a reduction in nonradiative recombination losses [37,38]. Based on this evidence, the DMF:NMP and DMSO formulations were selected to continue the study. Time-resolved photoluminescence (TRPL) measurements were performed (Figure 1d) and showed a significant and comparable photoluminescence lifetime for the DMF:NMP and DMSO samples, confirming the results of the PL measurement, hence, the low number of charge traps in the films from both samples [39]. The stability of the two samples in air was also tested by keeping the samples in ambient air for 7 and 14 days and recording the absorption spectra over time (Figure 1e,f). The samples were found to be stable, showing only a small variation in the absorbance for the DMF:NMP and DMSO samples, whereas lower quality FAPI perovskites can show signs of degradation within a day [35], in some cases within a few hours [40], suggesting a good stability of the material over time, which is a fundamental requirement in the path toward the possible commercialization of the perovskite material. In particular, we did not observe the formation of the δ-phase in the films, which would imply irreversible degradation of the perovskite material and a significant drop in device performance.

These results are in line with a previous report showing how the preparation in air can increase the stability of FAPI thanks to the formation of bonds with the oxygen in the air, which are able to block the formation of the δ-phase [41]. We investigated the morphology of the samples by acquiring scanning electron microscopy (SEM) images and found that they were slightly different for the two formulations (Figure 2). While both samples had a staircase-like structure, the grain size was visibly larger in the case of DMF:NMP. On the other hand, the morphology was more homogeneous in the DMSO sample as the grain sizes were smaller and close to an average value of 1.2 µm, whereas the grain size of the DMF:NMP sample ranged between 0.5 µm and 2.5 µm. It is worth mentioning that, apart from the different solvent systems, the two samples also differed in annealing temperature, where the DMF:NMP sample was annealed at 165 °C while the DMSO sample was annealed at 150 °C, which could explain the smaller grain size, as it has been shown that higher annealing temperatures promote the formation of larger grains [42]. Nevertheless, the temperature was optimized for the two formulations to achieve complete conversion of the perovskite precursors to perovskite. As a result of the characterizations, we evaluated the DMF:NMP and DMSO formulations as suitable for air deposition and decided to move onto the device fabrication. As a first step, the DMF:NMP formulation was selected for the initial optimization, since the larger grain size combined with good crystallinity holds promise for PSCs with good performances. Devices were fabricated in inverted p-i-n configuration, but we found that there were serious wettability issues with the FAPI ink on the HTL when deposited in ambient air. This is a known issue for the inverted p-i-n architecture [43], but the problem was exacerbated in ambient air, and particularly in high humidity environments (RH > 50%). To address this issue, different HTLs were tested in the device’s structure to determine the best one in terms of both wettability and performance. We also fabricated devices on both ITO and FTO as transparent conductive oxides (TCO), since while ITO is far more popular as a bottom electrode, FTO has a higher surface roughness, which could be beneficial for the adhesion of the perovskite solution [44]. The criteria used to select the HTLs for testing were the compatibility of the material with the perovskite solution, as hygroscopic materials combine better with the polar perovskite solution, and the promise of high efficiency according to the results reported in the literature. The selected HTLs were poly(bis{4-phenyl}{2,4,6trimethylphenyl}amine (PTAA), 2PACz and poly(3,4-ethylenedioxythiophene) polystyrene sulfonate (PEDOT:PSS), and the device structure and current–voltage (I–V) curves are shown in Figure 3a,b, respectively. PTAA and 2PACz were chosen for reaching high efficiency, while PEDOT:PSS was chosen primarily to increase the coverage since it is more hydrophilic, allowing for good spreading of the perovskite solution on the substrate. Among the selected HTLs, PTAA was the worst in terms of wettability and the deposition of a continuous FAPI perovskite was not feasible. This is not surprising given the hydrophobic nature of the PTAA layer, which can provide protection against moisture [45] but also reduces the adhesion of the polar perovskite solution. For this reason, a thin interlayer of 4-fluoro-phenethylammonium iodide (FPEAI) was deposited atop the PTAA HTL to improve the wettability and device performances [46]. SEM images of the perovskite samples deposited on the different HTLs are shown in Figure 3c. We found that even with the interlayer of FPEAI, the coverage on PTAA was really poor and several pinholes were visible in the SEM image. The incomplete coverage severely affected both the reproducibility of the deposition and the device performance (Figure 3c). In addition, the FAPI grains were slightly different and smaller compared to those of the FAPI deposited on 2PACz, which ultimately reduced the current density of the device [47]. The films on 2PACz and PEDOT:PSS were significantly better, although some pinholes were present in all cases. The best morphology was obtained with 2PACz as HTL, which favors the formation of large and compact grains. PEDOT:PSS, on the other hand, although it has a very good wettability due to the hygroscopic nature of the material [48], had the worst morphology, with very small and disordered grains. Accordingly, the performances were the worst, with very low Voc and Jsc values. Although it has been reported in several works that PEDOT:PSS as HTL in PSCs causes losses in the Voc value because the work function does not adequately match the work function of the perovskite [49,50], it is clear that the morphology of the perovskite contributes significantly to the device performance. In the final analysis, the self-assembly monolayer of 2PACz was found to be the best HTL, both in terms of morphology and performance, with maximum comparable power conversion efficiencies of 15.9% on ITO and 15.2% on FTO (Table 1). Specifically, while the morphologies were very similar, the devices with FTO/2PACz achieved the highest value of Voc, while the devices with ITO/2PACz had the highest value of Jsc (Figure 3b), which is consistent with other reports that ITO generally promotes a higher Jsc compared to FTO [51] in inverted perovskite solar cells. This comparison confirms the superiority of the SAMs as HTL, which not only allowed us to obtain a good coverage in humid ambient air with a good perovskite morphology, but, as was already reported, they are able to provide both fast extraction and passivation at the interface, significantly limiting the non-radiative losses resulting in high Voc values [52].

Once 2PACz was found to be the optimal HTL to achieve high efficiency and good wettability of the perovskite inks, devices were fabricated using the DMSO formulations and the performances were compared with the DMF:NMP devices. The device structure was ITO/2PACz/FAPI/PCBM/BCP and the performances are shown in Figure 4. ITO was chosen as the transparent conductive oxide instead of FTO as it gave the best overall efficiency. As can be clearly seen from the I–V curves and the photovoltaic parameters in Figure 4a,b, the DMSO formulation presented an enhanced average Voc value compared to the DMF:NMP formulation, suggesting the minimization of the recombination of the charge carriers and a perovskite material with fewer defects [53]. However, we also observed a decrease in the Jsc values, consistently with the lower dimension of the perovskite grains. As a result, the PCE values were slightly lower for the DMSO formulation, although still comparable to the DMF:NMP formulation, with a maximum value of 15.1% of efficiency. Finally, the stability of the photovoltaic devices was assessed through maximum power point tracking measurements under the continuous illumination of 100 (mW/cm^2^) in the glove box (Figure 4c). The device based on the DMF:NMP formulation showed a better stability, maintaining almost the performance, up to 96% of the efficiency after 20 h of illumination, while the device based on the DMSO formulation suffered a more significant power loss, retaining 90% of the initial efficiency after the same time. These results suggest the superiority of the DMF:NMP formulation for the air deposition of FAPI, both in terms of efficiency and stability. Nevertheless, the DMSO formulation was also proven to be suitable for ambient air deposition, achieving comparable efficiency and particularly high Voc values. Importantly, DMSO has a lower toxicity when compared to DMF [26], making it even more compatible for the possible upscaling of the PSC technology.

## 4. Conclusions

In summary, in this study, we have shown the importance of the choice of solvent in the fabrication of FAPI perovskite solar cells in ambient air, as it plays a fundamental role in determining the morphology and stability of the perovskite. Two formulations of FAPI perovskite based on DMF:NMP and DMSO were found to be ideal for the fabrication of PSCs in ambient air in the inverted p-i-n configuration, producing good quality perovskite films without the presence of the δ-phase and were stable in air for over two weeks. In comparison, the DMF:NMP formulation produced FAPI perovskite with high crystallinity and superior morphology with a larger grain size, while the DMSO formulation presented a reduced grain size but higher PL intensity value. We optimized the device structure in an inverted configuration using the DMF:NMP formulation and tested different HTLs to overcome the wettability problem that arises in high humidity environments. The SEM images revealed a compact and ordered morphology as well as large grain size obtained with 2PACz as the HTL, whereas PTAA caused incomplete coverage and several pinholes in the films. Finally, PEDOT:PSS, thanks to its hygroscopic nature, helped to obtain a good wettability, but the resulting morphology was disordered and the grain size was significantly smaller. As a result, the self-assembly monolayer 2PACz was the best option to obtain almost complete coverage of the FAPI film and at the same time achieve the best performances, thanks to the 2PACz–FAPI interface. Moreover, we demonstrated that the efficiencies of the devices fabricated on FTO and ITO with 2PACz as the HTL are comparable and that, while ITO promotes higher Jsc values, FTO balances out by increasing the Voc values. Finally, devices were fabricated using the DMSO formulation with 2PACz as the HTL and resulted in a slightly lower efficiency when compared to the DMF:NMP formulation and also had poorer stability. However, the DMSO formulation remains very interesting thanks to the low number of defects in the film and the high Voc achieved, which gives hope for higher efficiencies. Furthermore, the DMSO formulation allows us to avoid toxic solvents, which makes it a favorable solution for air deposition and larger-scale deposition techniques.

## Figures and Tables

**Figure 1 nanomaterials-14-00107-f001:**
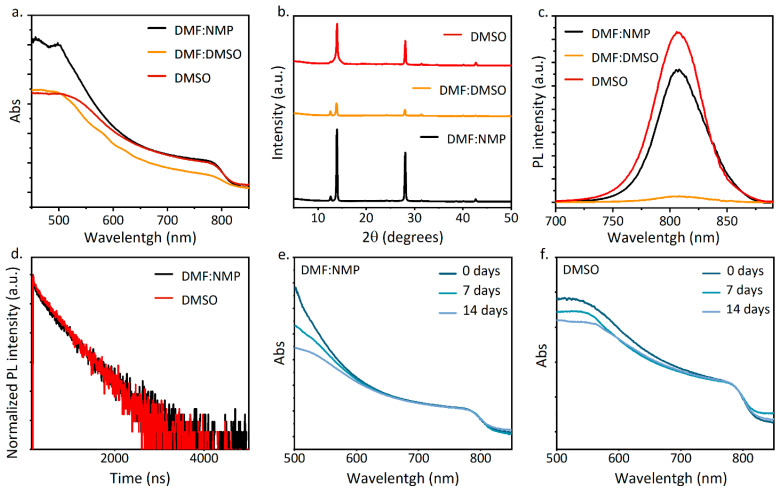
Optical and structural features of FAPI films deposited in ambient air. (**a**) UV–Vis absorption spectra. (**b**) XRD patterns. (**c**) Steady-state PL spectra. (**d**) PL lifetime spectra. (**e**) UV–Vis absorption spectra over time of the DMF:NMP sample. (**f**) UV–Vis absorption spectra over time of the DMSO sample.

**Figure 2 nanomaterials-14-00107-f002:**
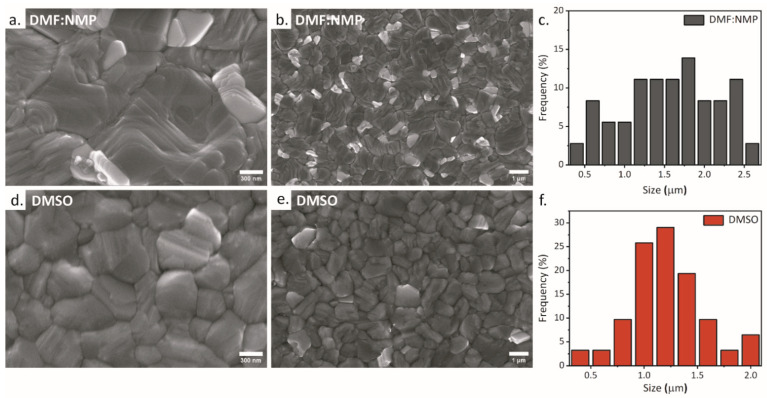
SEM top-view images of the FAPI perovskite films on 2PACz and the grain size distributions. (**a**–**c**) DMF:NMP samples. (**d**–**f**) DMSO samples.

**Figure 3 nanomaterials-14-00107-f003:**
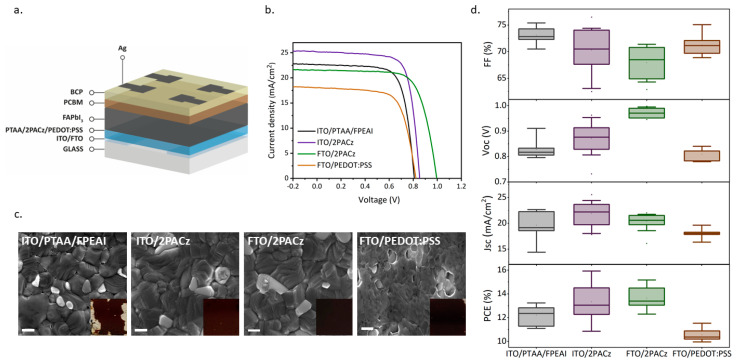
(**a**) Schematics of perovskite solar cell architecture. (**b**) I−V curves of FAPI DMF:NMP on: ITO/PTAA/FPEAI, ITO/2PACz, FTO/2PACz and FTO/PEDOT:PSS in reverse scan. (**c**) SEM images with a scale of 400 nm of FAPI DMF:NMP on the different HTLs coupled with photographic images of the sample. (**d**) The statistical photovoltaic parameters of FAPI DMF:NMP on: ITO/PTAA/FPEAI, ITO/2PACz, FTO/2PACz and FTO/PEDOT:PSS.

**Figure 4 nanomaterials-14-00107-f004:**
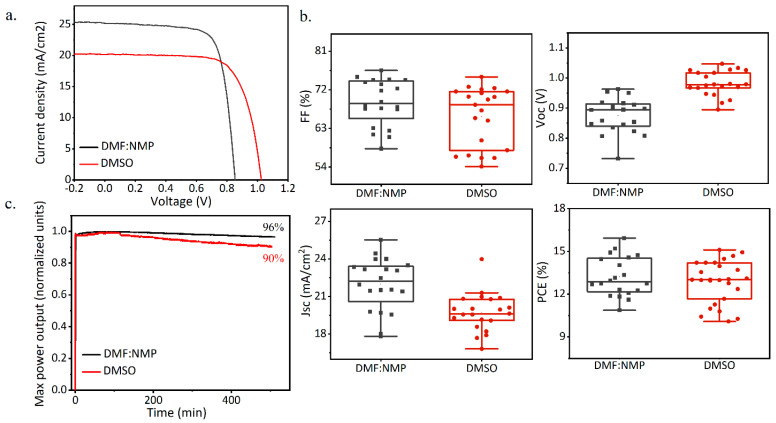
(**a**) I−V curves of FAPI DMF:NMP and FAPI DMSO on 2PACz in reverse scan. (**b**) The statistical photovoltaic parameters of FAPI DMF:NMP and FAPI DSMO. (**c**) Maximum power point tracking measures under continuous illumination of devices based on FAPI DMF:NMP and DMSO.

**Table 1 nanomaterials-14-00107-t001:** Photovoltaic parameter of FAPI DMF:NMP on: ITO/PTAA/FPEAI, ITO/2PACz, FTO/2PACz, and FTO/PEDOT:PSS.

	FF (%)	Voc (V)	Jsc (mA/cm^2^)	PCE (%)
ITO/PTAA/FPEAI	73.1 ± 1.6	0.83 ± 0.03	19.6 ± 2.8	12.2 ± 0.8
best device	72.5	0.809	22.61	13.3
ITO/2PACz	70.3 ± 4.0	0.87 ± 0.06	21.7 ± 2.2	13.3 ± 1.3
best device	73.8	0.845	25.53	15.9
FTO/2PACz	67.9 ± 3.6	0.97 ± 0.02	20.3 ± 1.6	13.6 ± 0.9
best device	70.8	0.995	21.54	15.2
FTO/PEDOT:PSS	71.2 ± 2.0	0.81 ± 0.02	18.1 ± 0.90	10.5 ± 0.5
best device	75.1	0.841	18.28	11.5

## Data Availability

Data are contained within the article.

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
