# Peer review of "Formamidinium Perovskite Deposition in Ambient Air Environment for Inverted p-i-n Solar Cells"

_nanomaterials, 2024, doi:10.3390/nano14010107_

Round 1

Reviewer 1 Report (Previous Reviewer 3)

Comments and Suggestions for Authors

The authors addressed all the concerns arose from the 1st review round, thus I recommend this paper for publication in Nanomaterials.

Comments on the Quality of English Language

Minor editing of English language required.

Author Response

We thank the reviewer for accepting our manuscript. 

Reviewer 2 Report (New Reviewer)

Comments and Suggestions for Authors

The authors investigated different formulations of pure FAPI for the fabrication of perovskite solar cells (PSCs) in air. They reported that formulations using a mixture of N,N-Dimethylformamide (DMF): N-Methyl-2-pyrrolidone (NMP) and only Dimethyl sulfoxide (DMSO) are suitable for the deposition in air. To fabricate inverted p-i-n solar cells, they tested different hole transporting layers (HTLs) and observed the effects on the wettability of the perovskite solution and on the performance. A self-assembly monolayer of 2PACz (2-(9H-Carbazol-9-yl)ethyl]phosphonic Acid), was found to be the best option as HTL, allowing us to achieve efficiencies >15% on both FTO and ITO. However, there are some problems with this manuscript. The paper could be further considered for publication with addressing the following issues:

1.      Why did the authors choose the different deposition conditions for each sample? Shouldn’t it be better to use the same deposition conditions for comparison?

2.      In line 157, the authors attribute the diffraction peak at 12.60° to the 6H hexagonal phase. Although, the 101 reflection of the 6H phase is around 12.2° and the 102 reflection should be overlapping with the 100 reflection of the alpha phase (~14°). In fact, the reflection at 12.60° is form PbI2 (same reference used in the manuscript: https://doi.org/10.1021/acsenergylett.7b00981). It is also visible in the SEM (Figure 2). Please review all the XRD peak assignments.

3.      There’s no mention about the shift on what would be the 100 and 200 peaks of the DMSO sample. The shift seems to be too high to be neglected. Also, the peak presents an enlargement, with could be attributed to different phases being present. Please discuss this part of the manuscript better.

4.      Why was the PL spectra cut at 850 nm? Also, the PL spectrum of DMF:NMP sample looks broader then DMSO sample. Please explain these differences in more detail.

5.      Please plot the y axis of the TRPL in log scale. It is impossible to see any difference when plotting on a linear scale.

6.      I would like to see some devices with DMSO sample, since it showed the higher PL and better TRPL, according to the authors. Just the small grain size shouldn’t justify not making devices with this sample. You should include this, at least, as supporting information.

7.      The authors should explore the stability in this work, as one of the motivations of making the perovskite in air conditions is its higher stability.

Minor:

1.      The Hexagonal phases are more commonly referred as 2H, 4H, 6H,…, not H2, H4, H6,…,as referred in this manuscript. Please review it.

2.      The XRD Diffraction is never a “spectrum”, as mentioned in the manuscript (line 157, for example. Spectrum refers to a plot where a signal is plotted as function of frequency. Please correct it. You can use XRD Diffraction pattern, for example.

Author Response

Comments to the Author

The authors investigated different formulations of pure FAPI for the fabrication of perovskite solar cells (PSCs) in air. They reported that formulations using a mixture of N,N-Dimethylformamide (DMF): N-Methyl-2-pyrrolidone (NMP) and only Dimethyl sulfoxide (DMSO) are suitable for the deposition in air. To fabricate inverted p-i-n solar cells, they tested different hole transporting layers (HTLs) and observed the effects on the wettability of the perovskite solution and on the performance. A self-assembly monolayer of 2PACz (2-(9H-Carbazol-9-yl)ethyl]phosphonic Acid), was found to be the best option as HTL, allowing us to achieve efficiencies >15% on both FTO and ITO. However, there are some problems with this manuscript. The paper could be further considered for publication with addressing the following issues:

  1. Why did the authors choose the different deposition conditions for each sample? Shouldn’t it be better to use the same deposition conditions for comparison?

Answer: We thank the referee for his attention to our work.

We have modified the deposition conditions to suit the different formulations. As the formulations have different molarity due to the different solubility of the precursors in the solvents, the spin-coating parameters are chosen to obtain good quality films with a similar thickness.

We add the following sentence to the revised manuscript to clarify this point:

“The deposition conditions are adjusted to suit the different formulations. The formulations have different molarity due to the different solubility of the precursors in the solvents, thus the spin-coating parameters are optimised to obtain good quality films with a similar thickness.”

  1. In line 157, the authors attribute the diffraction peak at 12.60° to the 6H hexagonal phase. Although, the 101 reflection of the 6H phase is around 12.2° and the 102 reflection should be overlapping with the 100 reflection of the alpha phase (~14°). In fact, the reflection at 12.60° is form PbI2 (same reference used in the manuscript: https://doi.org/10.1021/acsenergylett.7b00981). It is also visible in the SEM (Figure 2).

Please review all the XRD peak assignments.

Answer: Thank you for finding the inconsistency, we reviewed the XRD peak assignments.

  1. There’s no mention about the shift on what would be the 100 and 200 peaks of the DMSO The shift seems to be too high to be neglected. Also, the peak presents an enlargement, with could be attributed to different phases being present. Please discuss this part of the manuscript better.

Answer: We thank the referee for enlightening the issue, we found that there was a graphical error making up the shift in the DMSO pattern and a distortion in the height of the peaks. Regarding the broadening of the peak, we attributed it to the smaller grain size of the sample, as confirmed by the SEM images.

We add the following sentence to clarify this point:

“In comparison, the DMF:NMP sample has higher crystallinity as the (1 0 0) reflection is higher and sharper if compared to the (1 0 0) reflection in the DMSO sample, which shows a broadening, suggesting lower crystallinity and smaller grain size.”

  1. Why was the PL spectra cut at 850 nm? Also, the PL spectrum of DMF:NMP sample looks broader then DMSO sample. Please explain these differences in more detail.

Answer: We fixed the image expanding the X axis to 900 nm. Regarding the PL peak broadness, we calculated the full width at half maximum (FWHM) of the DMF:NMP and DMSO samples finding that they are equivalent.

  1. Please plot the y axis of the TRPL in log scale. It is impossible to see any difference when plotting on a linear scale.

Answer: We reported the TRPL in log scale.

  1. I would like to see some devices with DMSO sample, since it showed the higher PL and better TRPL, according to the authors. Just the small grain size shouldn’t justify not making devices with this sample. You should include this, at least, as supporting information.

Answer: We added a figure and a discussion on the devices with the DMSO formulation in comparison with the DMF:NMP devices in the revised manuscript.

  1. The authors should explore the stability in this work, as one of the motivations of making the perovskite in air conditions is its higher stability.

Answer: Regarding the stability, we reported a maximum power point measurement of the devices based on the DMF:NMP and DMSO formulations.

Minor:

  1. The Hexagonal phases are more commonly referred as 2H, 4H, 6H,…, not H2, H4, H6,…,as referred in this manuscript. Please review it.

Answer: We fixed the mistake.

  1. The XRD Diffraction is never a “spectrum”, as mentioned in the manuscript (line 157, for example. Spectrum refers to a plot where a signal is plotted as function of frequency. Please correct it. You can use XRD Diffraction pattern, for example.

Answer: We corrected the error.

Reviewer 3 Report (New Reviewer)

Comments and Suggestions for Authors

The work presented by Vanni et al. is indeed interesting, however, a few questions from my side arise: 

The introduction mentions the success of perovskite solar cells (PSCs) in terms of power conversion efficiency. Could you provide more context on the specific advancements or breakthroughs that have contributed to this success?

The paper highlights issues related to environmental control during the fabrication of perovskite layers. How critical is this issue, and how does it impact the scalability of manufacturing PSCs?

The paper uses three different formulations of FAPI with different solvent systems. Could you explain the rationale behind choosing these specific solvent systems, and what are the implications of using each in terms of film quality and device performance?

The selection of hole-transporting layers (HTLs) is discussed. What are the key considerations when choosing an HTL, and why was 2PACz chosen as the optimal solution?

The paper discusses the stability of FAPI films in ambient air over time. What are the potential implications of this stability for practical applications and long-term device performance?

The study compares the morphology and performance of devices on ITO and FTO with 2PACz as HTL. Could you elaborate on the reasons behind these choices and how the differences in morphology impact device performance?

The SEM images show differences in grain size and morphology for FAPI films on different HTLs. How do these differences correlate with the performance variations observed in the devices?

The stability of FAPI films in ambient air is tested over 7 and 14 days. What are the observed changes in the films over this period, and how do these changes relate to the overall stability of the devices?

The study concludes by emphasizing the suitability of two FAPI formulations for ambient air fabrication of PSCs. What are the key takeaways regarding the stability and performance of FAPI-based PSCs from this research?

Are there any potential limitations or challenges that the study identifies for future research in the field of perovskite solar cells?

Thanks a lot for answering these questions!

Comments on the Quality of English Language

Discrete

Author Response

Referee: 3

Comments to the Author

The work presented by Vanni et al. is indeed interesting, however, a few questions from my side arise:

  1. The introduction mentions the success of perovskite solar cells (PSCs) in terms of power conversion efficiency. Could you provide more context on the specific advancements or breakthroughs that have contributed to this success?

Answer: We thank the referee for the attention reserved to our work. We added a discussion in the introduction to extend the context around the perovskite advancements:

Starting from the first reported PSC with an efficiency of 10.9% in 20123, the PCEs have been increased to more than 25%4 in recent years thanks to the numerous methods to improve the efficiency and stability of the devices, many of which target the passivation of defects that causes ion migration and non-radiative recombination, which ultimately affect the performance and stability of the device5. Other strategies include interfacial modifications, acting on the optimisation of the charge transport materials6, and composition engineering7.”

  1. The paper highlights issues related to environmental control during the fabrication of perovskite layers. How critical is this issue, and how does it impact the scalability of manufacturing PSCs?

Answer: We have expanded the manuscript introduction to answer the question:

“Exposure to atmospheric moisture is known to be detrimental for metal halide perovskite materials. This limitation becomes more relevant when moving from lab scale spin-coater deposition to larger scale deposition methods, for which having a controlled environment can becomes quite expensive.”

  1. The paper uses three different formulations of FAPI with different solvent systems. Could you explain the rationale behind choosing these specific solvent systems, and what are the implications of using each in terms of film quality and device performance?

Answer: We have expanded the discussion on the solvent selection in revised manuscript to answer the question:

“The role of the solvent goes far beyond solubilising of the precursors, as it participates in the perovskite formation, in particular by controlling the growth of the nuclei and slowing down the reaction rate. Choosing the right solvent therefore becomes essential to achieving high performance PSC. We chose to test the DMF:DMSO and DMF:NMP formulations as they represent the ideal model of a non-ligand solvent (DMF) and a ligand solvent (DMSO or NMP) with a strong coordination ability to form a stable intermediate adduct, thus retarding the crystallisation and allowing for a better control and the production of highly uniform FAPI films. While DMSO is the most commonly used additive as Lewis base, it has been shown that NMP is a better option in the case of FA-based perovskites due to the stronger interaction between NMP and the FA cation. Moreover, the toxicity of the solvents needs also be considered, especially looking towards the possible upscaling of the PSC technology. Therefore, we also chose to test a formulation using only DMSO as solvent to eliminate highly toxic solvents.”

  1. The selection of hole-transporting layers (HTLs) is discussed. What are the key considerations when choosing an HTL, and why was 2PACz chosen as the optimal solution?

Answer: The key considerations for the choice of HTL were high efficiency and wettability of the perovskite solution. PTAA and 2PACz were chosen for reaching high efficiency, while PEDOT:PSS was chosen primary to increase the coverage since it is more hydrophilic. 2PACz was chosen as the optimal solution since it provides the highest efficiency combined with a good wettability of the ink.

We add the following paragraphs in the revised manuscript to answer this question:  

The criteria used to select the HTLs for the testing were the compatibility of the material with the perovskite solution, as hygroscopic materials combine better with the polar perovskite solution, and the promise of high efficiency according to the results reported in literature.”….

PTAA and 2PACz were chosen for reaching high efficiency, while PEDOT:PSS was chosen primary to increase the coverage since it is more hydrophilic allowing a good spreading of the perovskite solution on the substrate.”

  1. The paper discusses the stability of FAPI films in ambient air over time. What are the potential implications of this stability for practical applications and long-term device performance?

Answer: The stability in air over time is a fundamental prerequisite to move towards larger scale deposition methods and the commercialization of PSCs, since the formation of the delta phase would significate the irreversible degradation of the perovskite and a significant drop in device performance.

We add the following paragraph in the revised manuscript to answer this question:  

“The samples were found to be stable, showing only a small variation in the absorbance for the DMF:NMP and DMSO samples, whereas lower quality FAPI perovskites can show signs of degradation within a day, in some cases within a few hours, suggesting a good stability of the material over time which is a fundamental requirement in the path towards the possible commercialization of the perovskite material. In particular, we did not observe the formation of δ-phase in the films, which would imply irreversible degradation of the perovskite material and a significant drop in device performance.”

  1. The study compares the morphology and performance of devices on ITO and FTO with 2PACz as HTL. Could you elaborate on the reasons behind these choices and how the differences in morphology impact device performance?

Answer: We chose to test FTO as well as ITO because FTO has a higher surface roughness, resulting in a better wettability of the substrate. The morphology on ITO/2PACz and FTO/2PACz are practically the same and the changes in the device performance arise from the different properties of the conductive oxides.

  1. The SEM images show differences in grain size and morphology for FAPI films on different HTLs. How do these differences correlate with the performance variations observed in the devices?

Answer: The SEM images show how the compact and ordered morphology as well as large grain size obtained with 2PACz as HTL promote high efficiency devices, while PTAA causes incomplete coverage and several pinholes in the films. Finally, PEDOT:PSS, thanks to its hygroscopic nature, helps obtaining a good wettability but the resulting morphology is disordered and the grain size is significantly lower.

We add the following paragraph in the revised manuscript to answer this question: 

In comparison, the DMF:NMP formulation produces FAPI perovskite with high crystallinity and superior morphology with larger grain size, while the DMSO formulation presents reduced grain size but higher PL intensity value”

The SEM images revealed compact and ordered morphology as well as large grain size obtained with 2PACz as HTL, whereas PTAA causes incomplete coverage and several pinholes in the films. Finally, PEDOT:PSS, thanks to its hygroscopic nature, helps to obtain a good wettability but the resulting morphology is disordered, and the grain size is significantly smaller”

  1. The stability of FAPI films in ambient air is tested over 7 and 14 days. What are the observed changes in the films over this period, and how do these changes relate to the overall stability of the devices?

Answer: We did not observe any major changes in the absorbance spectra of the films in 14 days of air exposure, especially we did not find evidence of δ-phase formation in the films, which would ultimately impact the device performance by being not-photoactive.

  1. The study concludes by emphasizing the suitability of two FAPI formulations for ambient air fabrication of PSCs. What are the key takeaways regarding the stability and performance of FAPI-based PSCs from this research?

Answer: The study highlights the importance of solvent selection in the fabrication of FAPI perovskite solar cells, as it plays a fundamental role in the crystallisation process and affects the stability of the α-phase in the films. We found that the DMSO formulation, although giving slightly lower efficiency, is very promising as it is less toxic compared to DMF in view of possible upscaling of the PSC technology.

We add the following paragraph in the revised manuscript to answer this question:

“Finally, devices were fabricated using the DMSO formulation with 2PACz as HTL and resulted in a slightly lower efficiency if compared to the DMF:NMP formulation and also poorer stability. However, the DMSO formulation remains very interesting thanks to the low number of defects in the film and the high Voc achieved, which gives hope for higher efficiencies. Furthermore, the DMSO formulation allows to avoid toxic solvents, which makes it a favourable solution for air deposition and larger scale deposition techniques.”

  1. Are there any potential limitations or challenges that the study identifies for future research in the field of perovskite solar cells?

Answer: Wettability on the HTL remains a huge issue that must be addressed especially going towards large scale deposition where the perovskite crystallization process cannot be easily controlled.

“Furthermore, devices were fabricated with both of the formulations and different hole transporting layers (HTLs) were compared to address the wettability issue that arises in humid ambient air and represents a major challenge, especially for larger scale depositions.”

Round 2

Reviewer 2 Report (New Reviewer)

Comments and Suggestions for Authors

The authors addressed all my points. I, therefore, accept the manuscript for the publication in the present form.

Reviewer 3 Report (New Reviewer)

Comments and Suggestions for Authors

All good from my side

This manuscript is a resubmission of an earlier submission. The following is a list of the peer review reports and author responses from that submission.

Round 1

Reviewer 1 Report

Comments and Suggestions for Authors

The authors present interesting research on a new method of obtaining metal halide perovskite solar cells in ambient air. The work has enough original elements to be considered and is in the scope of the Journal. However, before I can recommend publication, the authors must perform a Major Revision, in which the following observations are addressed:

1.      The authors must outline the fact that this processing technique may also benefit the fabrication processes of other devices, such as perovskite and hybrid metasurfaces. I suggest the following papers as reference reading: https://doi.org/10.3390/ma16113934 , https://doi.org/10.3390/ma15175944 , https://doi.org/10.3390/nano9050791 ,  https://doi.org/10.1002/adom.202101120

2.      The IV curves of the sample devices must also be included and discussed in the paper. Without them, the values of FF, Voc, Jsc and PCE alone are not justified.

3.      There is no comparison to existing work, in order to highlight the pros and cons: The authors must authors compare the performance of the devices processed in an uncontrolled ambient atmosphere, to a reference FAPI cell built following the standard procedure, in an N2 atmosphere, or at least mention the reported state-of-the-art parameter values of a standard FAPI cell.

4.      Spell checks necessary throughout the paper.

Comments on the Quality of English Language

  Spell checks necessary throughout the paper.

Author Response

Referee: 1

Comments to the Author

The authors present interesting research on a new method of obtaining metal halide perovskite solar cells in ambient air. The work has enough original elements to be considered and is in the scope of the Journal. However, before I can recommend publication, the authors must perform a Major Revision, in which the following observations are addressed:

  1. The authors must outline the fact that this processing technique may also benefit the fabrication processes of other devices, such as perovskite and hybrid metasurfaces. I suggest the following papers as reference reading: https://doi.org/10.3390/ma16113934 , https://doi.org/10.3390/ma15175944 , https://doi.org/10.3390/nano9050791 , https://doi.org/10.1002/adom.202101120

Answer: We thank the referee for the attention reserved to our work. We changed the introduction of the manuscript pointing out that studying fabrication procedures in air can benefit other devices.

  1. The IV curves of the sample devices must also be included and discussed in the paper. Without them, the values of FF, Voc, Jsc and PCE alone are not justified.

Answer: We modified the draft which now includes the IV curves of the devices.

  1. There is no comparison to existing work, in order to highlight the pros and cons: The authors must compare the performance of the devices processed in an uncontrolled ambient atmosphere, to a reference FAPI cell built following the standard procedure, in an N2 atmosphere, or at least mention the reported state-of-the-art parameter values of a standard FAPI cell.

Answer: We provided the state-of-the-art PCE of the standard FAPI inverted p-i-n PSC fabricated in N2 controlled atmosphere in the introduction.

  1. Spell checks necessary throughout the paper.

Answer: An extensive editing of the English language was made.  

Reviewer 2 Report

Comments and Suggestions for Authors

Comments on the Quality of English Language

English may require moderate editing.

Author Response

Referee: 2

Comments to the Author

This paper presents the fabrication of formamidinium perovskite solar cell (FPSC) under the ambient environmental condition. Although FP has an ideal band gap, as the introduction says, its not stable at room temperature (see the first paragraph of introduction). My question is then how long such FPSCs are going to last? The authors have studied the stability for 7 to 14 days but is this adequate? I would like the authors to add a few sentences on the stability of FPSCs in the discussion and/or conclusions/abstract. Other than the above, I am happy with the presentation of results/ measurements, etc., and paper may be accepted if the above required statements are satisfactory.

Answer: We thank the referee for the suggestions. We added in the discussion the consideration that, even though 14 days is not a long time, the degradation in delta phase is often visible in a much less time, such as a day or even hours. In addition, we hinted in the conclusion that a more in-depth study is required to confirm the stability over longer periods of time and that it could starting point for an investigation on the differences between the two formulations.

Reviewer 3 Report

Comments and Suggestions for Authors

The authors coated the perovskite FAPI on various hole transport layers such as PTAA, 2PACz, and PEDOT:PSS using different solution-based techniques in an air ambient, and analyzed their surface morphologies as well as their effects on solar cell characteristics. After reading this paper, it was found that the present form of the paper does not provide adequate experimental data and their corresponding explanations and discussion, particularly, regarding the strong photoluminescence characteristics with a long photocarrier lifetime for the perovskite FAPI. Furthermore, no I-V characteristics of the inverted pin solar cells were presented in the paper. Overall, I do not recommend this paper for publication in Nanomaterials.

 Main comments in details are listed below.

1. The definitions of some abbreviations, such as PSC, DMSO, HTL, and PSC, are not correctly placed throughout the paper.

2. Even though FAPI was synthesized in air, detailed analyses and their corresponding explanations and discussion on the strong photoluminescence with a long photocarrier lifetime of air-synthesized FAPI (DMSO and DMF:NMP) were missed.

3. If Fig. 1(d) represents the time-resolved PL data, the x-axis should be corrected to time.

4. The caption of Fig. 1(f) should be corrected.

5. In the description of Fig. 1, it was explained that, in XRD, no delta-phase FAPI-related peaks were observed in all samples. However, in the absorption spectra, broad onsets and shoulders were explained as resulting from the presence of delta-phase within the film. An analysis related to peak broadening or delta-phase FAPI is required.

6. The experimental results that 2PACz was formed to be a self-assembly monolayer should be presented.

7. The I-V characteristics of the fabricated solar cell samples should be provided.

8. English should be checked carefully by native speakers or colleagues who are fluent in English.

Comments on the Quality of English Language

For better flow of the contents and the story line, moderate editing of English language required.

Author Response

Referee: 3

Comments to the Author

The authors coated the perovskite FAPI on various hole transport layers such as PTAA, 2PACz, and PEDOT:PSS using different solution-based techniques in an air ambient, and analyzed their surface morphologies as well as their effects on solar cell characteristics. After reading this paper, it was found that the present form of the paper does not provide adequate experimental data and their corresponding explanations and discussion, particularly, regarding the strong photoluminescence characteristics with a long photocarrier lifetime for the perovskite FAPI. Furthermore, no I-V characteristics of the inverted pin solar cells were presented in the paper. Overall, I do not recommend this paper for publication in Nanomaterials.

  Main comments in details are listed below.

The definitions of some abbreviations, such as PSC, DMSO, HTL, and PSC, are not correctly placed throughout the paper.

Answer: We thank the referee for detecting the mistake. We fixed the order of the definitions of the abbreviations.

Even though FAPI was synthesized in air, detailed analyses and their corresponding explanations and discussion on the strong photoluminescence with a long photocarrier lifetime of air-synthesized FAPI (DMSO and DMF:NMP) were missed.

Answer: We added a discussion on the increased photoluminescence and the long photoluminescence lifetime in the manuscript highlighting the higher quality of the DMF:NMP and DMSO samples pointing out the correlation between higher PL intensity and long lifetimes and lower number of nonradiative recombination losses.

If Fig. 1(d) represents the time-resolved PL data, the x-axis should be corrected to time.

Answer: We corrected the x-axis with the proper measurement unit.

The caption of Fig. 1(f) should be corrected.

Answer: We corrected the error in the caption of Fig.1(f)

In the description of Fig. 1, it was explained that, in XRD, no delta-phase FAPI-related peaks were observed in all samples. However, in the absorption spectra, broad onsets and shoulders were explained as resulting from the presence of delta-phase within the film. An analysis related to peak broadening or delta-phase FAPI is required.

Answer: We thank the referee for finding this inconsistency in the manuscript. After a more in-depth analysis we found that the XRD peak at 12.6°, which was attributed by us to PbI2, is caused by another hexagonal polymorph which is often mistaken for PbI2. This can explain the presence of the shoulders in the absorbance spectrum of the DMF:DMSO sample since the amount of hexagonal phase present in the sample is way more higher compared to the other samples, in which the cubic phase peak is predominant. The paper is now corrected accordingly.

The experimental results that 2PACz was formed to be a self-assembly monolayer should be presented.

Answer: The formation of the monolayer of 2PACz was achieved following the same protocol already reported in literature and the details are reported in the experimental section. In particular, we made sure that the molecules in excess were removed through a washing step of the substrate after the 2PACz deposition. For more clarity, we added a citation for the procedure to obtain the 2PACz monolayer.  

The I-V characteristics of the fabricated solar cell samples should be provided.

Answer: We included the I-V curves in the draft.

  1. English should be checked carefully by native speakers or colleagues who are fluent in English.

Comments on the Quality of English Language

For better flow of the contents and the story line, moderate editing of English language required.

Answer: An extensive editing of the English language was made.